# The E3 Ubiquitin Ligase NEDD4-1 Mediates Temozolomide-Resistant Glioblastoma through PTEN Attenuation and Redox Imbalance in Nrf2–HO-1 Axis

**DOI:** 10.3390/ijms221910247

**Published:** 2021-09-23

**Authors:** Hao-Yu Chuang, Li-Yun Hsu, Chih-Ming Pan, Narpati Wesa Pikatan, Vijesh Kumar Yadav, Iat-Hang Fong, Chao-Hsuan Chen, Chi-Tai Yeh, Shao-Chih Chiu

**Affiliations:** 1School of Medicine, China Medical University, Taichung 40447, Taiwan; greeberg1975@gmail.com; 2Translational Cell Therapy Center, Tainan Municipal An-Nan Hospital-China Medical University, Tainan 70967, Taiwan; 3Division of Neurosurgery, Tainan Municipal An-Nan Hospital-China Medical University, Tainan 70967, Taiwan; 4Division of Neurosurgery, China Medical University Beigang Hospital, Beigang Township 65152, Taiwan; 5Department of Emergency Medicine, Shuang-Ho Hospital-Taipei Medical University, New Taipei City 23561, Taiwan; 17003@s.tmu.edu.tw; 6Graduate Institute of Injury Prevention and Control, Taipei Medical University, Taipei 110, Taiwan; 7Department of Emergency Medicine, School of Medicine, Taipei Medical University, Taipei 110, Taiwan; 8Translational Cell Therapy Center, Department of Medical Research, China Medical University Hospital, Taichung 40447, Taiwan; jim0427@gmail.com; 9Doctorate Program of Medical and Health Science, Faculty of Medicine, Public Health and Nursing, Universitas Gadjah Mada, Yogyakarta 55281, Indonesia; narpatiwp@gmail.com; 10Department of Medical Research and Education, Taipei Medical University-Shuang Ho Hospital, New Taipei City 235, Taiwan; vijeshp2@gmail.com (V.K.Y.); 18149@s.tmu.edu.tw (I.-H.F.); 11Biomedicine Institution, Department of Neurosurgery, China Medical University, Taichung 40447, Taiwan; d13407@mail.cmuh.org.tw; 12Department of Medical Laboratory Science and Biotechnology, Yuanpei University of Medical Technology, Hsinchu 300, Taiwan; 13Graduate Institute of Biomedical Sciences, China Medical University, Taichung 40447, Taiwan; 14Drug Development Center, China Medical University, Taichung 40447, Taiwan

**Keywords:** NEDD4-1, ubiquitin ligase, glioblastoma, TMZ resistance, indole-3-carbinol

## Abstract

Background: Glioblastoma (GBM) is the most common primary malignant brain tumor in adults. It is highly resistant to chemotherapy, and tumor recurrence is common. Neuronal precursor cell-expressed developmentally downregulated 4-1 (NEDD4-1) is an E3 ligase that controls embryonic development and animal growth. NEDD4-1 regulates the tumor suppressor phosphatase and tensin homolog (PTEN), one of the major regulators of the PI3K/AKT/mTOR signaling axis, as well as the response to oxidative stress. Methods: The expression levels of NEDD4-1 in GBM tissues and different cell lines were determined by quantitative real-time polymerase chain reaction and immunohistochemistry. In vitro and in vivo assays were performed to explore the biological effects of NEDD4-1 on GBM cells. Temozolomide (TMZ)-resistant U87MG and U251 cell lines were specifically established to determine NEDD4-1 upregulation and its effects on the tumorigenicity of GBM cells. Subsequently, miRNA expression in TMZ-resistant cell lines was investigated to determine the dysregulated miRNA underlying the overexpression of NEDD4-1. Indole-3-carbinol (I3C) was used to inhibit NEDD4-1 activity, and its effect on chemoresistance to TMZ was verified. Results: NEDD4-1 was significantly overexpressed in the GBM and TMZ-resistant cells and clinical samples. NEDD4-1 was demonstrated to be a key oncoprotein associated with TMZ resistance, inducing oncogenicity and tumorigenesis of TMZ-resistant GBM cells compared with TMZ-responsive cells. Mechanistically, TMZ-resistant cells exhibited dysregulated expression of miR-3129-5p and miR-199b-3p, resulting in the induced NEDD4-1 mRNA-expression level. The upregulation of NEDD4-1 attenuated PTEN expression and promoted the AKT/NRF2/HO-1 oxidative stress signaling axis, which in turn conferred amplified defense against reactive oxygen species (ROS) and eventually higher resistance against TMZ treatment. The combination treatment of I3C, a known inhibitor of NEDD4-1, with TMZ resulted in a synergistic effect and re-sensitized TMZ-resistant tumor cells both in vitro and in vivo. Conclusions: These findings demonstrate the critical role of NEDD4-1 in regulating the redox imbalance in TMZ-resistant GBM cells via the degradation of PTEN and the upregulation of the AKT/NRF2/HO-1 signaling pathway. Targeting this regulatory axis may help eliminate TMZ-resistant glioblastoma.

## 1. Introduction

Glioblastoma (GBM) is the most common primary malignant brain tumor in adults, with a median survival time of 14 months following diagnosis [1]. Chemotherapy is one of the standard therapeutic methods for reducing tumor size, inhibiting distant metastasis, and extending patient survival. However, GBM is highly resistant to chemotherapy, and tumor recurrence is common [2]. Recent studies have demonstrated a close correlation between chemotherapy resistance and the antioxidant response system [3,4]. Chemotherapy induces cell death by inducing DNA damage through the generation of reactive oxygen species (ROS). ROS are highly reactive molecules formed by living organisms as a result of normal cellular metabolism and environmental influence, and they damage nucleic acids, lipids, and proteins, altering their functions. Inflammatory cells, mitochondria, and peroxisomes are all endogenous sources of ROS [5]. Low levels of ROS may act as a signal transducer, whereas higher levels of ROS may induce cell death [6]. Thus, redox balance regulation may serve as an interesting target in enhancing the effectiveness of chemotherapy or overcoming drug resistance in cancers.

A subset of cancer cells possesses enhanced proliferation and survival abilities in addition to their self-renewal ability and oncogenic transforming properties. These cells termed cancer stem cells (CSCs) also possess an enhanced antioxidant response system, conferring high resilience to oxidative stress and apoptosis to tumors, which may even be enhanced by ROS [7,8]. Notably, CSCs are also less sensitive to radiotherapy and many currently available chemotherapy regimens [9]. These cells acquire the aforementioned properties during malignant progression by reactivating a complex process called epithelial-to-mesenchymal transition (EMT), which is integral in embryonic development, wound healing, and CSCs behavior [10]. CSCs are maintained by a core group of primary transcription factors that are influenced by a wide variety of developmental signals and extracellular cues [11]. Among these cues, the PI3K/AKT/mTOR signaling pathway is one of the main perpetrators that maintains and enhances the population of CSCs in various cancers [12,13,14]. Thus, through this study, we investigated the role of neuronal precursor cell-expressed developmentally downregulated 4-1 (NEDD4-1), a founding member of the NEDD4 family of E3 ubiquitin ligase, known for its ubiquitination activity on molecule phosphatase and tensin homolog (PTEN), one of the main regulators of the PI3K/AKT/mTOR signaling axis [15,16,17].

E3 ubiquitin ligases (E3s) have been the subject of extensive studies in recent years. Several studies have reported that E3s are strongly linked to tumorigenesis, metastasis, and prognosis [15,18,19,20]. Ubiquitination, a post-translational modification, is a highly organized sequence of enzymatic reactions involving E1, E2, and E3s, which target proteins for degradation or cause other cellular fates, such as regulating enzymatic activity; inflammatory signaling, endocytosis, and histone modification together with ubiquitination have been implicated in various cancers [21,22]. Substrate proteins flagged by Ub are degraded, activated, or transported by the Ub-proteasome system following various forms of ubiquitination [22,23]. NEDD4-1, also known as NEDD4 and RPF1, was first isolated in 1992 from mouse neural precursor cells whose mRNA levels were downregulated during the growth of the mouse brain. Because of its widespread expression in the placenta, liver, thyroid, skin, endometrium, gall bladder, urinary bladder, and kidney, NEDD4-1 may play a role in various human cellular functions [24]. According to subsequent research, NEDD4-1 is an E3 ligase that controls embryonic development and animal growth. NEDD4-1 has several upstream and downstream genes, and, with its perceived dual role in cancer, it can be used as a molecular switch to control tumor growth through its competitive substrates [15,18,24,25]. To our knowledge, this is the first study on the novel role of NEDD4-1, an E3 ligase, in GBM TMZ resistance.

Importantly, in this study, we found that NEDD4-1 was significantly upregulated in GBM, and that the transcriptional activation of endogenous NEDD4-1 degraded PTEN expression and enabled the NRF2/HO-1 antioxidant signaling response, which in turn conferred GBM cells with resistance to temozolomide (TMZ) chemotherapy. We also elucidated the role of dysregulated miRNAs in the population of GBM CSCs, which subsequently triggered the overexpression of NEDD4-1 expression. Our findings describe, at least partially, the role and mechanism of NEDD4-1 in conferring drug chemoresistance through the attenuation of PTEN and the subsequent hyperactivity of the AKT/NRF2/HO-1 signaling cascade in GBM.

## 2. Results

### 2.1. NEDD4-1 Was Overexpressed in GBM Tissue Samples

To assess the potential role of NEDD4-1 in the development of human GBM, we first investigated the expression of NEDD4-1 by utilizing the large publicly available dataset in the R2 database. The expression level of NEDD4-1 at the mRNA level across three GBM databases comprising normal brain tissues was evaluated. NEDD4-1 at the mRNA level was significantly overexpressed in GBM tissues as compared to its normal counterpart (Figure 1A). Furthermore, we validated this finding using the TCGA-GBM database; the results demonstrated NEDD4-1 was more highly expressed in tumor tissue than in normal tissue (Figure 1B). Consistently, our in-house SHH-GBM patient cohort tumor IHC staining showed an increased expression level of NEDD4-1 in the GBM tissue compared with normal tissue (Figure 1C,D). Further, using Kaplan–Meier analysis, we demonstrated that higher NEDD4-1 expression conferred a worse prognosis to patients with GBM patients (Figure 1E,F). These findings advocate that NEDD4-1 overexpression is associated with the development of human GBM.

### 2.2. TMZ-Resistant GBM Cell Lines Had an Increased Endogenous Expression of NEDD4-1

For our next experiments, we aimed to determine the role of NEDD4-1 in TMZ-resistant GBM cells. To mimic patients with higher expression of NEDD4-1 in the clinical setting, we measured the endogenous expression of NEDD4-1 protein in a panel of GBM cell lines using Western blotting and selected the cell lines with the highest expression of NEDD4-1, namely, U251 and U87MG (Figure 2A). We proceeded to establish a TMZ-resistant cell line by incubating the cell lines with a low dose of TMZ, as described in Section 4. As demonstrated in the cell viability assay in Figure 2B, U87MGR and U251R cell lines were more resistant to TMZ treatment than U87MG and U251 cell lines. We also observed that these TMZ-resistant cells possessed a higher tendency to form tumorspheres than their TMZ-responsive counterparts (Figure 2C). These tumorspheres also showed an increased endogenous expression level of NEDD4-1 compared with adherent U87MG and U251 cell lines (Figure 2D). Next, we incubated 1 × 10^3^ U87MG, U251, U87MGR, and U251R cells in six-well plates for 14 days. The cells were then stained with crystal violet and analyzed. The results demonstrated that TMZ-resistant cell lines have induced clonogenicity potential compared to TMZ-responsive cells (Figure 2E). This series of experiments helped us to establish the oncogenicity of TMZ-resistant GBM cells.

### 2.3. Dysregulation of miR-3129-5p and miR-199b-3p Consequently Leads to an Induced Expression of NEDD4-1 in TMZ-Resistant GBM Cells

To further understand the role of NEDD4-1 expression in GBM cells, especially in TMZ-resistant cells, we speculated that dysregulated microRNAs could account for the upregulation of NEDD4-1 in such resistant cells. Thus, we utilized the TargetScan database and procured the top 10 microRNAs that are most strongly correlated with NEDD4-1 enlisted in Table 1. Next, the expression of these 10 miRNAs was compared in TMZ-responsive U87 cells with TMZ-resistant U87 cells using qRT-PCR analysis. As shown in Figure 3A, two miRNAs, miR-3129-5p and miR-199b-3p, were significantly downregulated among the 10 surveyed miRNAs in TMZ-resistant cells (Figure 3A). We subsequently procured both miR-3129-5p and miR-199b-3p mimics to validate NEDD4-1 as the gene target of these miRNAs. As NEDD4-1 has been previously shown to be associated with cell proliferation, and as its inhibition in our study affected the viability of GBM cells, we evaluated the staining intensity of the proliferation marker Ki-67 through immunofluorescence. As expected, single transfection of miR-3129-5p or miR-199b-3p mimics negatively affected Ki-67 staining intensity in the U87R cell line, whereas transfection of both mimics more strongly attenuated Ki-67 staining in the U87R cell line (Figure 3B). Next, we transfected U87R cells with the mimics of miR-3129-5p and/or miR-199b-3p and probed the cells for NEDD4-1 and PTEN expression through immunofluorescence. As expected, a single miRNA mimic for either miR-3129-5p or miR-199b-3p reduced the expression of NEDD4-1, whereas combining both mimics further reduced the expression of NEDD4-1 and increased the expression of PTEN (Figure 3C). Next, we transfected U87R cells with the wild-type (WT) or mutated (mut) NEDD4-1 3′untranslated region (3′UTR)-directed luciferase reporter and co-transfected the cells with miR-3129-5p and/or miR-199b-3p mimics. The result showed that WT-NEDD4-1-3′UTR luciferase activity was repressed by co-transfection of miRNA mimics, whereas mut-NEDD4-1-3′UTR luciferase activity was not significantly affected by these miRNA mimics, further confirming NEDD4-1 as the key target gene of both miR-3129-5p and miR-199b-3p (Figure 3D). Western blot analysis also followed the aforementioned trend, with a significant reduction in the expression of NEDD4-1 and an induced expression of PTEN with the transfection of either miR-3129-5p or miR-199b-3p mimics, whereas, with both, a strong and significant reduction in the expression of NEDD4-1 and an increased expression of PTEN were observed (Figure 3E). These results demonstrated that the NEDD4-1 upregulation in TMZ-resistant-GBM could be due to the dysregulated expression of miR-3129-5p and miR-199b-3p.

### 2.4. I3C Targets NEDD4-1 and Modulates the Oncogenic and Metastatic Phenotypes of TMZ-Resistant GBM Cells

I3C, a natural NEDD4 and WWP1 inhibitor derived from the family Brassicaceae, has been previously shown to effectively inhibit NEDD4-1 activity [26] (Figure 4A). Hence, we utilized this compound in our subsequent experiments to further elucidate the role of NEDD4-1 in GBM and to use this compound for translational purposes. Interestingly, we observed that TMZ-resistant U251R and U87R cell lines, as well as the parental (U251 and U87) cells, were sensitive to this compound (Figure 4B). I3C also effectively abrogated the tumorsphere-forming (Figure 4C), migratory (Figure 4D), and invasive (Figure 4E) abilities of U251R and U87R cell lines. Using Western blotting, we demonstrated that I3C treatment significantly reduced NEDD4-1 expression, together with a reduction in the expression of stemness markers (NESTIN and SOX2) and EMT (N-cadherin and vimentin) markers (Figure 4F), reflecting its anti-self-renewal, migration, and invasion properties in GBM cells.

### 2.5. I3C Increased Susceptibility of TMZ-Resistant Cells to TMZ by Inhibiting NEDD4-1-Induced PTEN Ubiquitination and Subsequently Inhibiting NRF2/HO-1 Antioxidant Signaling Response

Chemotherapies commonly work by straining cancer cells with oxidative stress and eventually killing them. Resistant cells may combat this by upregulating the defensive response against ROS. Thus, we investigated the role of I3C in dismantling this defense mechanism by inhibiting NRF2/HO-1 through NEDD4-1. We induced oxidative stress in the U87R cell line with *tert*-butyl hydroperoxide (TBHp) and treated these cells with/without I3C. TBHp is a well-known reactive oxygen species (ROC)-producing model substance for oxidative stress generation. TBHp-induced oxidative stress significantly increased NEDD4-1 expression, which led to a decreased protein level of PTEN and significantly higher AKT signaling activity, compared with the control. As expected, the NRF2/HO-1 signaling cascade was also strongly activated in the cells induced with TBHp. I3C treatment effectively inhibited NEDD4-1 and reverted the PTEN/AKT/NRF2/HO-1 signaling pathway (Figure 5A). We also demonstrated that I3C may reduce the expression of the antioxidant enzymes superoxide dismutase, catalase, and glutathione (Figure 5B). 2’,7’-Dichlorofluorescein diacetate (DCFDA) staining is the most commonly used probe to measure the accumulation of ROS in cells; through DCFDA staining, we demonstrated higher ROS accumulation in the cells treated with I3C (Figure 5C). Next, the potential of combining I3C treatment with TMZ was assayed. Using annexin V/propidium iodide staining, we demonstrated that the addition of I3C to TMZ treatment increased the number of positively stained cells, reflecting the increased apoptosis rate, compared with only TMZ treatment (Figure 5D). We ultimately combined the treatment of various doses of TMZ and I3C and observed a synergistic effect (isobologram, CI <1) of this drug combination on TMZ-resistant GBM cells (Figure 5E).

### 2.6. I3C Enhanced the Chemosensitivity of Xenografted TMZ-Resistant U87 Cell Line to TMZ

Next, we extrapolated our in vitro findings to in vivo experiments. A total of 1 × 10^6^ U87R cells were injected into the flank region of NOD/SCID mice. After the tumors were palpable, the mice were randomly grouped into the vehicle, TMZ, I3C, and TMZ + I3C treatment groups, with each group containing *n* = 10 mice. The tumors were weighed every 3 days. After completion of the in vivo experiment, the mice were humanely sacrificed on day 22, and the tumor tissues were extracted. In vivo experiment results demonstrated that I3C treatment could further sensitize the tumor to TMZ treatment (Figure 6A). However, we found no significant difference in the body weight of mice in the different groups, implying the nontoxic effect of I3C on mice (Figure 6B). We then evaluated the tumor tissue using the TUNEL assay, and the results revealed that tumors treated with a combination of I3C and TMZ showed an increased rate of TUNEL staining, indicating the increased apoptotic rate (Figure 6C). This series of experiments showed that I3C may help overcome chemoresistance to TMZ in GBM.

## 3. Discussion

In this study, we determined the role of the E3 ubiquitin ligase NEDD4-1 in GBM. Data analysis revealed that NEDD4-1 was overexpressed in GBM tumors compared with normal tissues. These data were strengthened by our in-house SHH-GBM clinical data, which showed that GBM tumors had a higher NEDD4-1 level than their normal tissue counterparts, as demonstrated by reverse transcription quantitative polymerase chain reaction (RT-qPCR) and IHC. In addition, at least two datasets showed that higher NEDD4-1 expression confers a worse prognosis in patients with GBM. Next, using TMZ-resistant GBM cell lines, we demonstrated that NEDD4-1 was highly expressed in such GBM cells compared with wild-type cells. We hypothesize that the abnormal expression of NEDD4-1 in GBM is due to dysregulation of the expression of miRNAs, which is often reported in chemoresistant cells [27,28].

miRNAs are a small class of noncoding RNA molecules (21–23 nucleotide fragments) that alter the expression of target mRNA, by pairing with complementary sequences within the 3′UTRs of targeted transcripts. miRNA has been shown to play a role in various biological processes, including cell growth, differentiation, apoptosis, and proliferation [29]. We generated a list of potential miRNAs that may be strongly associated with NEDD4-1 using TargetScan, a web-based database used to predict miRNA targets [30]. From 10 predicted miRNAs targeting NEDD4-1, we determined the two most downregulated miRNAs in the TMZ-resistant GBM cell line, namely, hsa-miR-3129-5p and hsa-miR-199b-3p. These two miRNAs significantly altered the NEDD4-1 level in the U87R cell line, as validated using their mimics and by further analyzing NEDD4-1 and PTEN levels using a combination of immunofluorescence, luciferase reporter, and Western blot assays. This series of experiments provided evidence regarding the involvement of miRNA in regulating NEDD4-1-induced PTEN ubiquitination in chemoresistant GBM cells.

Previous studies have also demonstrated the complexity of dysregulated miRNA expression involved in CSC characteristics, which results in GBM TMZ chemoresistance. Li et al. demonstrated that miR-186 was dysregulated in GBM-initiating cells (GICs); furthermore, the exogenous exposure of miR-186 mimic inhibited the GIC proliferation and reversed the resistance of GBM cells to cisplatin [31]. Sana et al. identified deregulated miRNA signatures in GIC clusters compared with non-stem-cell clusters. They found that these miRNAs were also associated with the poor survival of patients with GBM [32]. Thus, it is not surprising that NEDD4-1 is abnormally expressed in TMZ-resistant cell lines, considering that chemoresistant cells may have different biomolecular structures and functions, as reported in previous studies, including ours.

To subvert NEDD4-1 activity, we subsequently blocked its activity using a readily available E3 ubiquitin ligase inhibitor, I3C. I3C is extracted from glucobrassicin, which is a hydrolysis product of cruciferous vegetables such as cabbage, broccoli, and Brussels sprouts. I3C is a naturally occurring indole carbinol phytochemical implicated in various antiproliferative pathways in leukemia, melanoma, and breast, kidney, liver, colon, and cervical cancer [26]. Through this study, we demonstrated that both TMZ-resistant cell lines were sensitive to I3C treatment, and the addition of I3C attenuated the migration, invasion, and tumorsphere-forming potential of U87R and U251R cell lines. I3C treatment significantly reduced NEDD4-1 expression and subsequently reduced the stemness and EMT markers of TMZ-resistant GBM cell lines. Aronchik et al. reported that I3C suppressed in vivo tumor growth and induced PTEN protein expression levels in residual tumors in TMZ-responsive PTEN-expressing melanoma xenografts developed in athymic mice [33]. Our results contribute to the evidence of the role of I3C in reducing the tumorigenicity of TMZ-resistant GBM cells by targeting NEDD4-1.

Oxidative stress has long been linked with cancer chemoresistance. Cancer cells have a high basal level of ROS, rendering them more vulnerable than regular cells to an increase in ROS; chemoresistant cancer cells upregulate their antioxidant systems to become strongly adapted to intrinsic or drug-induced oxidative stress. In 2011, Olivia et al. specifically focused on the role of ROS in TMZ resistance of glioma cells. They reported that remodeling of the whole electron transport chain is linked to chemoresistance to TMZ, with large increases in the activity of complexes II/III and cytochrome c oxidase. They also found that glioma cells treated with antioxidants prevented TMZ toxicity. Furthermore, they demonstrated that TMZ-resistant mitochondrial DNA-depleted cells (rho degrees) had lower intracellular ROS levels after TMZ exposure than parental cells did [34]. In breast cancer cells, a robust proteomics approach using label-free mass spectrometry in patients who were responsive and resistant to chemotherapy revealed oxidative stress as one of the pivotal players of breast cancer chemoresistance [35]. Similarly, we demonstrated that the TMZ-resistant cell line is more resilient against TBHp-induced oxidative stress compared with chemoresponsive cells. Interestingly, IC3 treatment reversed the chemoresistance capacity of TMZ-resistant cells by attenuating NEDD4-1-induced degradation of PTEN, which ultimately inhibited the AKT/NRF2/HO-1 signaling pathway. Stripped of their defense mechanism, U87R cells were more susceptible to TMZ treatment. Our in vitro and in vivo data also showed that both IC3 and TMZ work synergistically to eliminate TMZ-resistant GBM cells.

## 4. Materials and Methods

### 4.1. Clinical Samples

We analyzed the microarray gene expression dataset of patients with GBM obtained from the GEO datasets through the R2 Genomics Analysis and Visualization Platform containing normal mixed male and female samples (Berchtold’s, GSE11882, *n* = 172 and Harris’s, GSE13564, *n* = 44), and tumor samples (Pfister, GSE36245, *n* = 46, Loeffler, GSE53733, *n* = 70, and Hegi, GSE7696, *n* = 84). Similarly, TCGA-GBM data, *n* = 173, were also applied. The R2 Genomics Analysis and Visualization Platform (https://hgserver1.amc.nl/cgi-bin/r2/main.cgi (accessed on 2 July 2020) was used to generate a Kaplan–Meier survival curve [36]. Parallel surgical tissues were collected and used to prepare formalin-fixed paraffin-embedded (FFPE) specimens. The China Medical University Hospital Research Ethics Committees approved the study (project approval number CMUH-109-REC2-014). The GBM tissues and normal tissues of all mixed male and female patients were collected; tissue neighboring the tumor tissue that was more than 3 cm away was considered normal. The clinicopathological information of the patients is described in Table 2**.** Tissue microarrays of GBMs were subjected to immunohistochemical (IHC) analysis after incubation with an antibody against NEDD4-1 (1:100 dilution, SC-81159; Santa Cruz Biotechnology, Inc., CA, USA) at 4 °C overnight. Staining using horseradish peroxidase (HRP), 3,3′-diaminobenzidine (dark brown), and hematoxylin (deep blue/purple) was performed according to the standard IHC protocol, followed by imaging and evaluation of the protein expression.

### 4.2. Cell Culture and Generation of Temozolomide-Resistant Cell Line

LN-229, T98G, U251, and U87MG GBM cell lines were obtained from the American Type Culture Collection (ATCC; Manassas, VA, USA) and were maintained in an incubator with 5% CO_2_ in humidified air. The cells were cultured in Dulbecco’s modified Eagle’s medium (#12491023; GIBCO, Life Technologies Corp., Carlsbad, CA, USA) supplemented with 10% fetal bovine serum (GIBCO, Life Technologies Corp.), penicillin (100 IU/mL), and streptomycin (100 g/mL) (#15140122, GIBCO, Life Technologies). TMZ-resistant U87MGR and U251R cells were generated as per the protocol suggested by Akiyama et al. (2014) [37]. The U87 and U251 parental cell lines, which are sensitive to TMZ, were first maintained in low doses of TMZ (5 µM) and then successively exposed to incremental doses of TMZ (up to 150 µM). After the killing of a majority of the cells, the surviving cells were maintained until a normal rate of growth was obtained. The TMZ-resistant cells were then maintained at a dose of 100 µM TMZ for in vitro and in vivo experiments. The IC_50_ value of TMZ was evaluated using the WST-1 assay. 

### 4.3. Western Blotting

After all treatments, GBM cells were harvested by trypsinization and lysed to extract the proteins using RIPA buffer (Cell Signaling Technology, Danvers, MA, USA) with a cocktail of protease inhibitors (Sigma Aldrich, St. Louis, MO, USA). Next, 10 μg of protein samples were separated using 10% SDS-PAGE electrophoresis and transferred to polyvinylidene fluoride (PVDF) membranes using the Bio-Rad Mini-Protein electro-transfer system (Bio-Rad, Taipei City, Taiwan). Immunoblotting was performed after blocking with 5% skimmed milk in Tris-buffered saline with Tween 20 (TBST) for 1 h and then incubated overnight at 4 °C with primary antibodies against the protein of interest (Appendix A). After incubation with the primary antibody, polyvinylidene difluoride (PVDF) membranes were washed thrice with TBST, incubated for 1 h at room temperature with an HRP-labeled secondary antibody, and rewashed with TBST. Subsequently, enhanced chemiluminescence, Western blotting reagents, and a BioSpectrum Imaging System (UVP; Upland, CA, USA) were used to detect the bands.

### 4.4. Transient Oligonucleotide Transfection

MiR-3129-5p and miR-199b-3p (mimics) and negative control miRNA were purchased from ThermoFisher Scientific (Taipei, Taiwan) and prepared under strict adherence to the vendor’s instructions. GBM cells were transfected using Lipofectamine^®^ 2000 (Invitrogen ThermoFisher Scientific, Inc., Waltham, MA, USA).

### 4.5. RNA Isolation and Reverse Transcription Quantitative Polymerase Chain Reaction

Total RNA was extracted from GBM and normal brain tissues using TRIzol™ reagents (Invitrogen; Thermo-Fisher Scientific, Inc., Waltham, MA, USA). Reverse transcription quantitative polymerase chain reaction (RT-PCR) was used to detect the expression of NEDD4-1 mRNA in GBM tissues. The NEDD4-1 primer sequences used were as follows: forward, 5′–CAGAAGAGGCAGCTTACAAGCC–3′; reverse, 5′–CTTCCCAACCTGGTGGTAATCC–3′. Glyceraldehyde-3-phosphate dehydrogenase (GAPDH) was considered as an internal reference to detect the NEDD4-1 mRNA expression level in the cells. All the selected mRNAs were pre-degenerated for 5 min at 95 °C and 1 min at 94 °C for 35 cycles. Then, they were pre-degenerated for 1 min at 56 °C, 2 min at 72 °C, and 10 min at 72 °C. The relative expression level of mRNA calculated through the 2^−∆∆Cq^ method (2^−∆∆Cq^ ≥ 2 was regarded as high expression).

### 4.6. Cell Sensitivity Assay/Clonogenic Assay

The “gold standard” cellular sensitivity/clonogenic assay was used to assess the sensitivities of GBM cells after the drug treatment. The cells were subcultured, seeded into six-well plates (2.7 × 10^4^ cells per well), and incubated at 37 °C for 2 days in 5% CO_2_. Cells were then cultured for an additional 24 h in medium fortified with 10% serum. The treated cells were then subcultured, reseeded at a concentration of 350 cells per well into a new six-well plate, and incubated for an additional 10 days at 37 °C in a 5% CO_2_ humidified incubator. The cells were dried after being set and were stained with 0.1% crystal violet. The experiments were conducted in triplicate.

### 4.7. Sulforhodamine B (SRB) Assay

The cell lines were grown in DMEM with 10% fetal bovine serum and 2 mM l-glutamine; 96-well microtiter plates were used to inoculate the cells. After inoculation and before the addition of compounds to be screened, the microtiter plates were incubated for 24 h at 37 °C under 5% CO_2_, 95% air, and 100% relative humidity. The cells were fixed in place by gently adding 50 mL of cold 10% *w*/*v* trichloroacetic acid, with incubation for 60 min at 4 °C. Subsequently, 50 µL of 0.4% *w*/*v* SRB solution in 1% CH_3_COOH was added to each well and incubated for 20 min at room temperature. Unbound dye was recovered after staining, and residual dye was removed by washing the well plates thoroughly with 1% CH_3_COOH and air drying. The bound stain was dissolved in a 10 mM Trizma base, and the absorbance was measured at 515 nm on an ELISA plate reader (690 nm reference wavelength).

### 4.8. Immunofluorescence Assay

U251R cells were cultured on glass coverslips before being transfected, as mentioned previously. After incubation, the cells were fixed for 15 min at 4 °C with 4% formaldehyde, permeabilized for 5 min with 0.01% Triton X-100, and blocked for 30 min at room temperature with 1% bovine serum albumin. The cells were then incubated for 24 h at 4 °C with the primary antibodies GSK-3 (#12456, 1:100, Cell Signaling Technology, Danvers, MA, USA) and β-catenin (#8814, 1:100, Cell Signaling Technology). Subsequently, the cells were stained with isotype-specific secondary antibody (Alexa Fluor^®^ 594-AffiniPure Donkey Anti-Rabbit IgG) for 1 h the following day (Jackson ImmunoResearch, West Grove, PA, USA).

### 4.9. Measurement of ROS Production

In a 96-well plate, the cells were plated at 20,000 cells per well in a final volume of 80 µL of the medium. Dichlorodihydrofluorescein diacetate (DCFDA; 10 µL of 50 µM) was applied to each well and incubated for 30 min. Fluorescence was measured on a microplate fluorometer (Tecan; Seestrasse, Männedorf, Switzerland), with an excitation filter set at 488 nm and an emission filter set at 530 nm.

### 4.10. Tumor Xenograft Study

For this study, 4–6-week-old female nonobese diabetic/severe combined immune-deficient (NOD/SCID) mice (mean weight, 17.4 ± 2.1 g) were purchased from BioLASCO (BioLASCO Taiwan, Taipei, Taiwan). The mice were inoculated subcutaneously with 2 × 10^6^ U251R cells in their hind flanks and were randomly assigned to vehicle (*n* = 10), TMZ (*n* = 10), indole-3-carbinol (I3C; *n* = 10), or TMZ + I3C (*n* = 10) groups. TMZ (2 mg/kg) treatment was initiated on day 8 when tumors started becoming palpable; TMZ was administered intraperitoneally (i.p.) every 72 h over the following 12 days. Using vernier calipers, tumor sizes were measured on days 6, 9, 12, 15, and 18 after GBM cell inoculation, and tumor volumes (v) were calculated as length (l) × (width (w))^2^ × 0.5. At the end of the experiment on day 18, the tumor-bearing mice were carefully sacrificed, and the tumors were extracted, examined, photographed, and measured again. All mice were housed under specific pathogen-free conditions and used following the animal care guidelines from the Institutional Animal Care and Use Committee of China Medical University Hospital (CMUIACUC-2020-150). The committee approved the experimental protocol.

### 4.11. Statistical Analysis

Means and standard errors of the mean (SEM) were used to present all the results. For multiple comparisons or repeated measurements, Student’s *t*-test was used. For multiple comparisons or repeated measurements, ANOVA or repeated ANOVA accompanied by Tukey’s post hoc test was used. Statistical significance was described as a *p*-value < 0.05. GraphPad Prism 7.0 was used to conduct statistical analysis (GraphPad Software, San Diego, CA, USA).

## 5. Conclusions

In summary, we described the critical role of NEDD4-1 in regulating the redox imbalance in TMZ-resistant GBM cells through the degradation of PTEN, the regulator of the AKT/NRF2/HO-1 signaling pathway. As described in our graphical abstract (Figure 7), TMZ-resistant cells underwent molecular changes that resulted in the deregulation of certain miRNAs, including miRNAs responsible for the expression of NEDD4-1. The resulting overexpression of NEDD4-1 is responsible for the tumorigenicity of GBM and the antioxidant response system that ultimately resists TMZ toxicity. Targeting NEDD4-1, particularly by using its direct inhibitor IC3, significantly reversed PTEN degradation, inhibited AKT/NRF2/HO-1 signaling, and increased ROS level and cell death in GBM cells. Thus, through this study, we partially describe the potential of targeting NEDD4-1 to reverse the resistance of GBM cells to TMZ. 

## Figures and Tables

**Figure 1 ijms-22-10247-f001:**
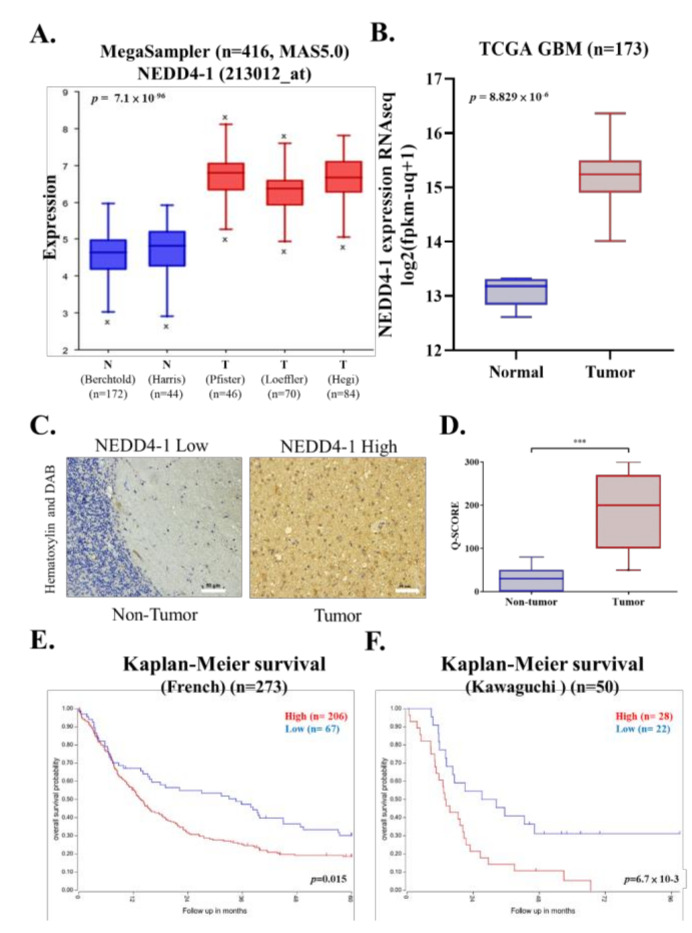
NEDD4-1 is associated with worse prognosis in glioblastoma. (**A**) NEDD4-1 expression evaluated across normal and glioblastoma datasets. (**B**) NEDD4-1 expression levels were evaluated from TCGA-GBM samples are grouped based on tumor and non-tumor counterparts. (**C**) Representative immunohistochemistry staining of clinical glioblastoma patient tissues. (**D**) Q-Score value comparison between non-tumor and tumor part taken from glioblastoma clinical tissue samples. Kaplan-Meier survival analysis of glioblastoma patients expressing NEDD4-1 was determined using gene correlation analysis by R2: Genomics Analysis and Visualization Platform (http://r2.amc.nl (accessed on 2 July 2021)) of the (**E**) French (*n* = 273) and (**F**) Kawaguchi (*n* = 50) datasets. *** *p* < 0.001.

**Figure 2 ijms-22-10247-f002:**
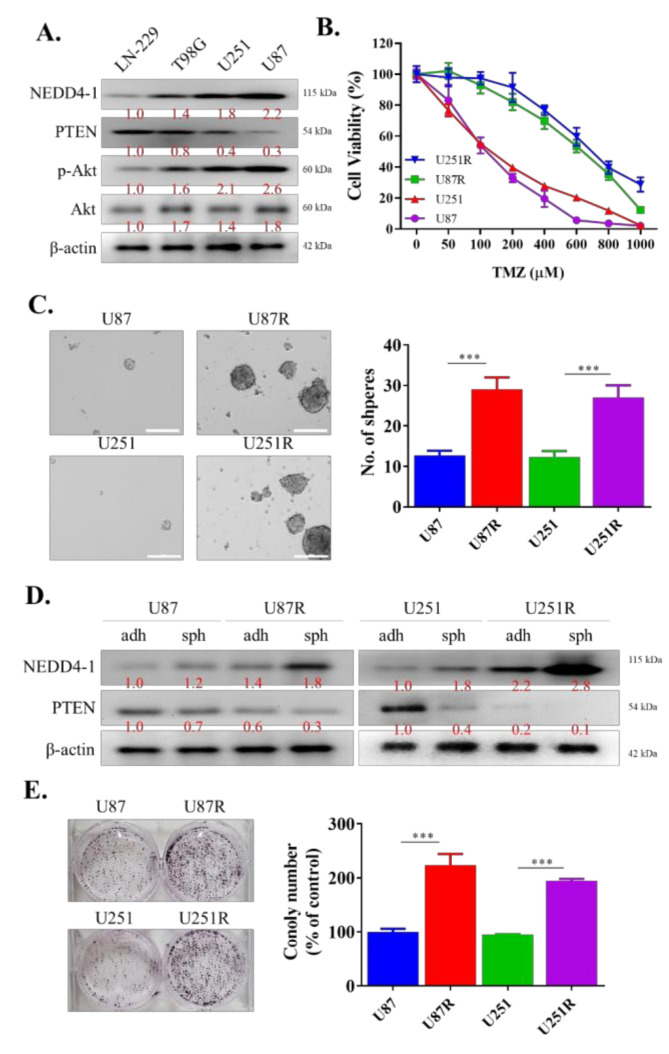
NEDD4-1 is highly expressed in temozolomide-resistant patient-derived GBM cell lines. (**A**) NEDD4-1/PTEN/AKT signaling pathway protein expression survey in glioblastoma cell lines. (**B**) Viability assay performed using SRB assay showing established TMZ-resistant cell lines, U87MGR and U251R, increased resilience against temozolomide treatment. (**C**) Tumorsphere formation assay demonstrating the higher potential of TMZ-resistant cell lines to form tumorspheres compared to their chemo-responsive counterpart. (**D**) Differential expression of NEDD4-1 and PTEN expression in adherent (adh) and tumorsphere (sph) cells in TMZ-responsive and TMZ-resistant U87MG and U251 cell lines. (**E**) Colony formation assay depicting higher clonogenicity of TMZ-resistant cell lines compared to their TMZ-responsive counterpart. Graph bars are mean ± SEM of 3 independent experiments. *** *p* < 0.001.

**Figure 3 ijms-22-10247-f003:**
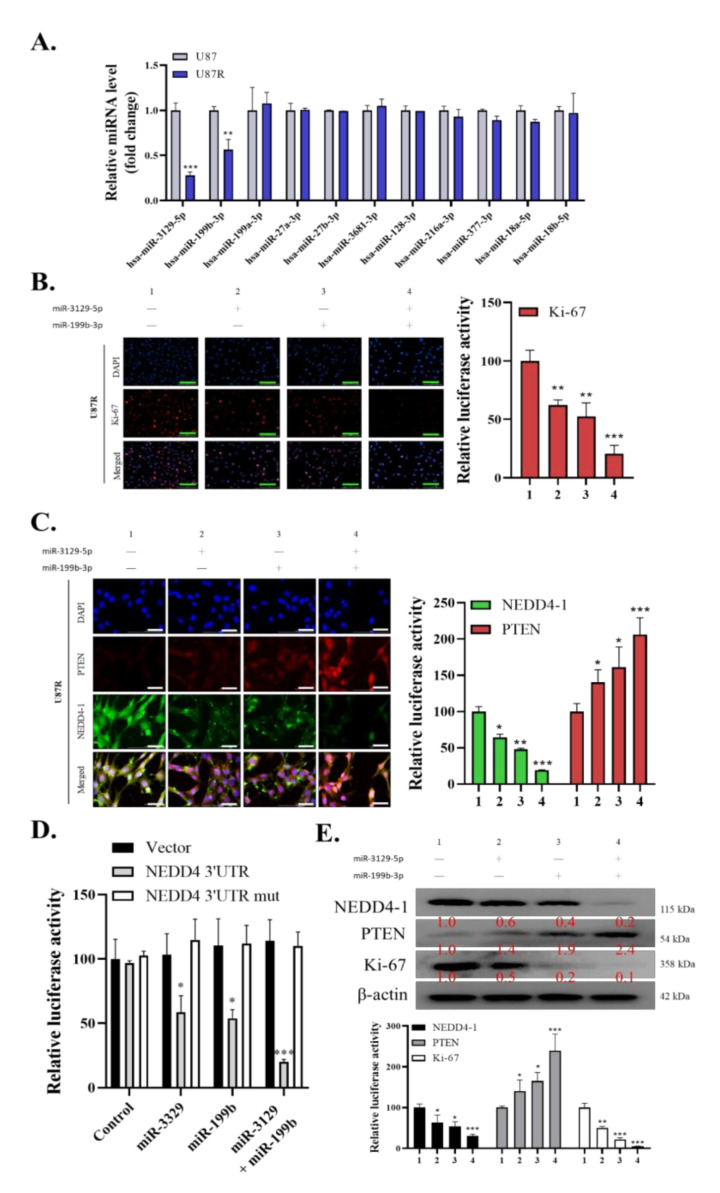
miR-3129-5p and miR-199b-3p regulate NEDD4-1 expression in GBM cell line. (**A**) The top 10 miRNAs with strongest correlation to NEDD4-1 according to the Targetscan database were evaluated for their expression level in U87MG and U87MG R cell lines to find the two most downregulated miRNAs. (**B**) Evaluation of Ki-67 immunofluorescence staining intensity of U87MG R cell line transfected with miR-3129-5p and miR-199b-3p mimics. DAPI staining (blue) was used to label DNA in all cells. Scale bar = 100 µm (**C**) Immunofluorescence staining of NEDD4-1 and PTEN in U87MGR cell line transfected with miR-3129-5p and miR-199b-3p mimics demonstrating differential localization. DAPI staining (blue) was used to label DNA in all cells. Scale bar = 100 µm. (**D**) U87MGR cells were co-transfected with miR-3129-5p, miR-199b-3p, or a control mimic (miR-NC), as well as a wild-type or mutated (mut) NEDD4-1 3′UTR-directed luciferase reporter. Luciferase activity was measured by dual-luciferase reporter assays. (**E**) Western blot analysis of NEDD4-1 and PTEN differential protein expression in U87MGR cells transfected with control mimic (miR-NC), miR-3129-5p, miR-199b-3p, or both. Graph bars are the mean ± SEM of three independent experiments. * *p* < 0.05, ** *p* < 0.01, *** *p* < 0.001.

**Figure 4 ijms-22-10247-f004:**
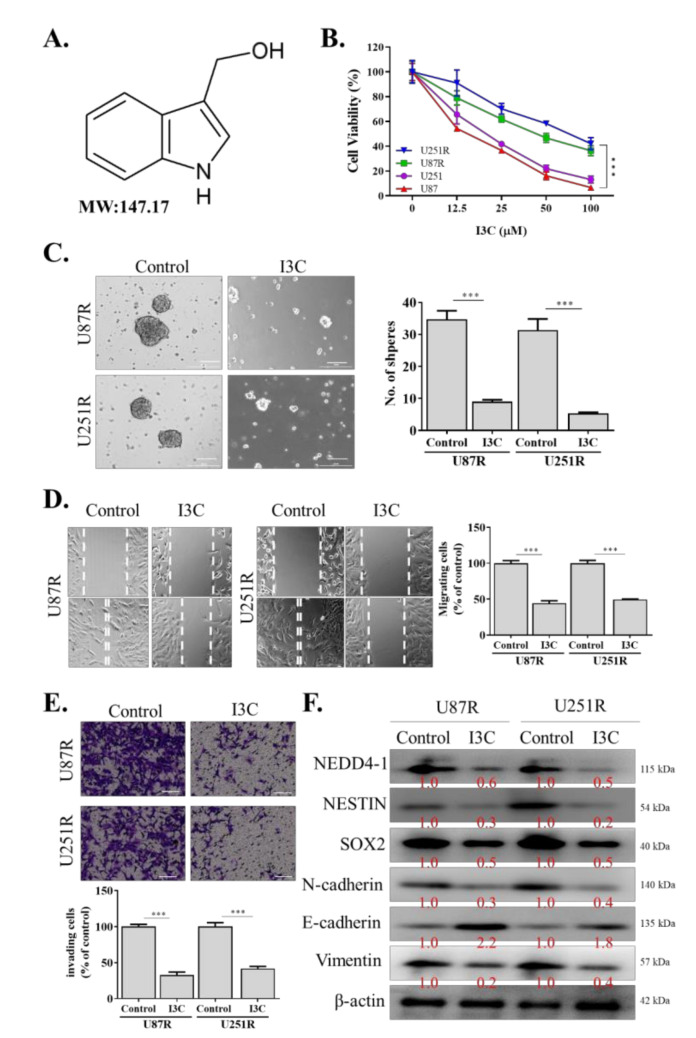
Effect of NEDD4-1 inhibitor, indole-3-carbinol (I3C), on migration, invasion, and tumorsphere formation of temozolomide-resistant GBM cell lines. (**A**) Molecular structure of indole-3-carbinol (I3C). (**B**) SRB cell viability assay of temozolomide resistant and parental cell lines treated with I3C. (**C**) I3C treatment attenuated U251R and U87MGR tumorsphere-forming potential. (**D**) U251R and U87MGR migration ability evaluated using scratch assay. Gap closure was evaluated after 24 h incubation. (**E**) Gel-coated Transwell chambers were utilized to evaluate U251R and U87MGR invasion ability after treatment with I3C for 24 h. (**F**) Protein expression of NEDD4-1, NESTIN, SOX2, N-cadherin, and vimentin after treatment with I3C for 24 h, evaluated using Western blot. Graph bars are the mean ± SEM of three independent experiments. *** *p* < 0.001.

**Figure 5 ijms-22-10247-f005:**
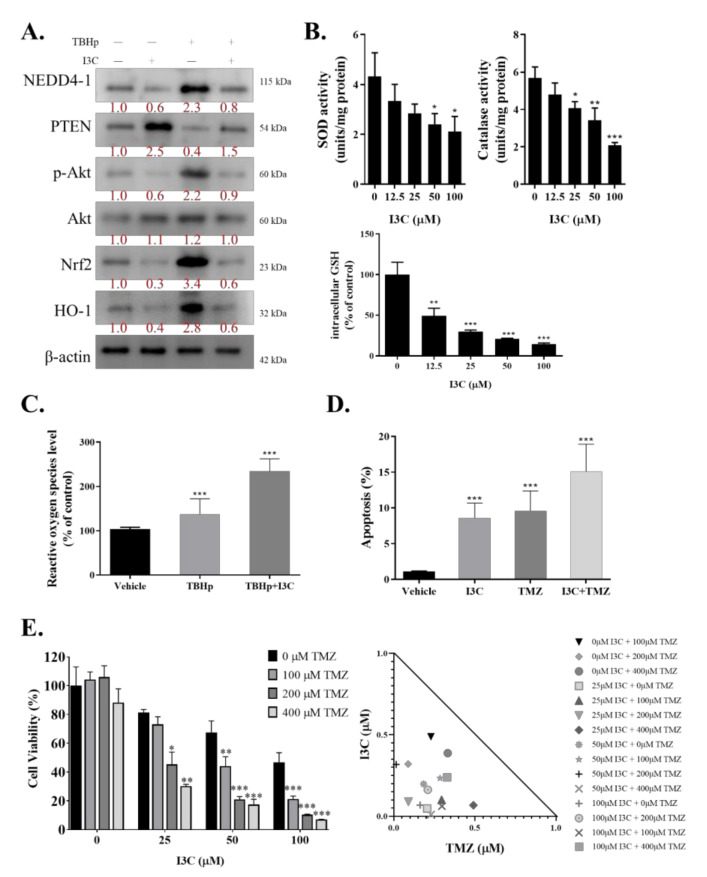
I3C inhibits NEDD4-1 expression and represses temozolomide-resistant GBM cell line Nrf2-induced antioxidant response. (**A**) Protein expression of NEDD4-1, PTEN, p-AKT, AKT, NRF2, and HO-1 was evaluated after inducing oxidative stress with TBHp and treatment with I3C. (**B**) Effect of various doses of I3C on antioxidant enzyme activity in U87MGR cells. (**C**) Flow cytometry analysis of TBHp-induced ROS formation in U87MGR cells with/without I3C treatment using the ROS-sensitive fluorometric probe DCFDA. Relative ROS production in all cells tested normalized to TBHp + vehicle group. (**D**) Cell apoptosis rate assay using annexin V/PI staining of U87MGR cells treated with I3C and/or temozolomide. (**E**) Cell viability assay of U87MGR cells treated with I3C and/or temozolomide. Graph bars are the mean ± SEM of three independent experiments. * *p* < 0.05, ** *p* < 0.01, *** *p* < 0.001.

**Figure 6 ijms-22-10247-f006:**
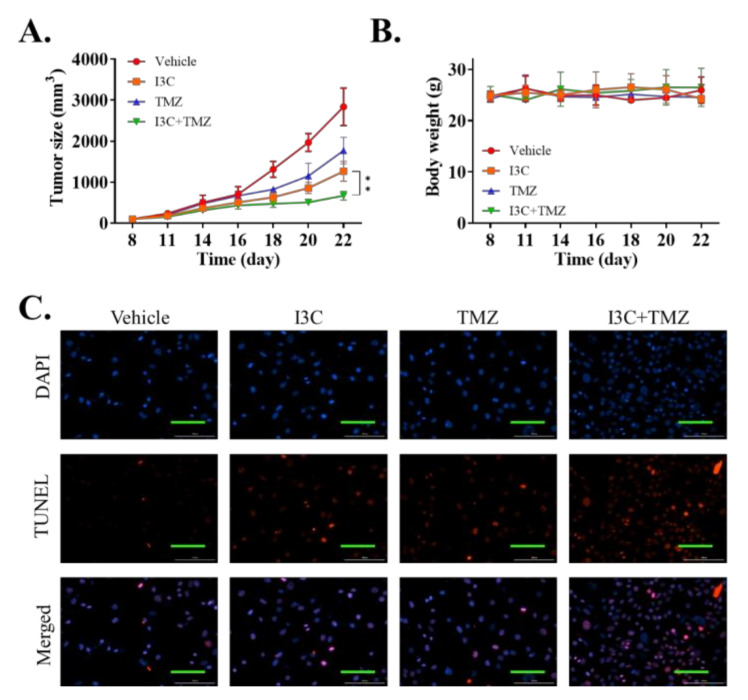
In vivo experiments demonstrating the effectiveness of I3C and temozolomide combination against xenograft temozolomide-resistant GBM tumor. (**A**) Tumor volume of mice xenografted with U87MGR cells with vehicle, temozolomide, I3C, or both drugs combined was evaluated every 3 days after day 8 of tumor implantation to evaluate the effectiveness of cisplatin treatment. (**B**) Mouse body weight was evaluated every 3 days. (**C**) Representative images of the TUNEL assay (red fluorescence of apoptotic cells and blue fluorescence of cell nuclei) were detected using a fluorescence microscope (×400). ** *p* < 0.01.

**Figure 7 ijms-22-10247-f007:**
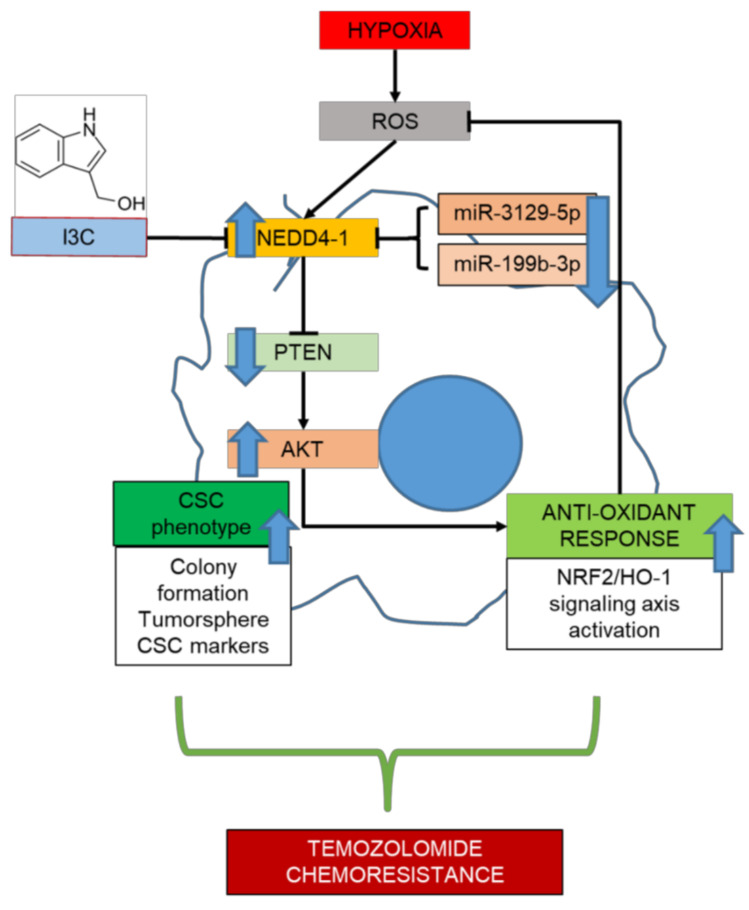
Graphical abstract. TMZ-resistant cells exhibited a dysregulated expression level of miR-3129-5p and miR-199b-3p, which results in the aberrant expression of NEDD4-1. Up-regulated NADD4-1 modulate the expression of tumor-suppressor PTEN and promotes the AKT/NRF2/HO-1 oxidative stress signaling axis on TMZ-resistant GBM cells. Thus, a combination of I3C, a well-known inhibitor of NEDD4-1 with TMZ results in the synergistic effect in improving therapeutics outcomes on TMZ-resistant GBM patients.

**Table 1 ijms-22-10247-t001:** Ten miRNAs with the strongest association with NEDD4-1 according to Targetscan database.

miRNA	Position in the UTR	Seed 1234match	Context++ Score	Context++ Score Percentile	Weighted Context++ Score	Conserved Branch Length	Pct
hsa-miR-3129-5p	366–373	8mer	−0.31	97	−0.31	7.3	0.87
hsa-miR-199b-3p	366–373	8mer	−0.34	96	−0.34	7.3	0.87
hsa-miR-199a-3p	366–373	8mer	−0.34	96	−0.34	7.3	0.87
hsa-miR-27a-3p	381–388	8mer	−0.43	99	−0.43	6.942	0.96
hsa-miR-27b-3p	381–388	8mer	−0.43	99	−0.43	6.942	0.96
hsa-miR-3681-3p	381–387	7mer-1A	−0.22	93	−0.22	6.942	0.77
hsa-miR-128-3p	381–387	7mer-1A	−0.17	90	−0.17	6.942	0.77
hsa-miR-216a-3p	381–387	7mer-1A	−0.15	88	−0.15	6.942	0.77
hsa-miR-377-3p	408–415	8mer	−0.11	72	−0.11	2.883	N/A
hsa-miR-18a-5p	538–545	8mer	−0.35	97	−0.33	5.828	0.83
hsa-miR-18b-5p	538–545	8mer	−0.35	97	−0.33	5.828	0.83

**Table 2 ijms-22-10247-t002:** Patients’ clinicopathological information.

Total no. Patients of Glioblastoma (*n* = 58)	No. (%)
**Age**	
<65	40 (68.4%)
≥65	18 (31.6%)
**Gender**	
male	34 (58.6%)
female	24 (41.4%)
**IDH1 status**	
wildtype	56 (96.6%)
mutation	2 (3.4%)
**Treatment modalities**	
Chemotherapy	24 (41.4%)
Radiotherapy	34 (58.6%)

## Data Availability

The datasets used and analyzed in the current study are publicly accessible, as indicated in the manuscript.

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
