# Peer review of "The E3 Ubiquitin Ligase NEDD4-1 Mediates Temozolomide-Resistant Glioblastoma through PTEN Attenuation and Redox Imbalance in Nrf2–HO-1 Axis"

_ijms, 2021, doi:10.3390/ijms221910247_

Round 1

Reviewer 1 Report

Chung et al. present a well-written manuscript which introduces NEDD4-1 mediated PTEN degradation in TMZ-resistant glioma. They found two specific microRNAs which down-regulates in TMZ-resistant glioma cell line and these microRNAs are essential for sustaining PTEN expression via down-regulation of NEDD4-1. In clinical aspect, they have shown that (1) the correlation between NEDD4 over-expression and worse prognosis in gliomablastoma, (2) I3C-induced NEDD4-1 down-regulation enhance TMZ-therapeutic efficacy. There are no major comment.

minor comments:

  1. line 117-119. If this manuscript is first report for NEDD4-1 over-expression in GBM, the author should describe it.
  2. line 139-140,148-149,160-161,181-182191-192,199-200,204-205,217-218. The author should insert spaces.
  3. Figure 1C. The author should explain histological color.
  4. Figure 1D. The author should describe P value mean of ***.
  5. Figure 1E and 1F. Half of the bottom of P values disappear.
  6. Figure 2A and 2D right. Can the author clear the background?
  7. Figure 2 legend. [resisttant] is miss-spelling.
  8. Figure 4D. The readers will not be able to find cells in this photo. Can the author modulate color balance or draw the line of migrated-tip cells?
  9. Figure 4F NESTIN,SOX2. Can the author clear the background?
  10. line 337,342. The author should simply explain TBHp and DCFDA.
  11. Figure 4B. U251 and U87 are lower expression of NEDD4-1 than each U251R and U87R. Are these cells higher cell viability upon I3C treatment?
  12. Figure 6B. Y axis is not "tumor size". Please change to "Body size".
  13. line 385-387. Please check the grammer.
  14. line 399. How mechanism does result in dysregulated miRNA? This concern may be future work and please discuss it if you are possible. Moreover, Dai et al. (Cancer Res.70.2951) have shown that FoxM1B transcription factor is overexpressed in human glioma tissues and upregulated NEDD4-1. Is there no alteration of FoxM1B expression in your glioma cell line?

Author Response

We thank the reviewer for carefully reading our manuscript and providing valuable comments. We accordingly response the questions raised by the Reviewer as follows:

Point-by-point responses to reviewer’s comments:

Dear Reviewer,

Coauthors and I very much appreciated the encouraging, critical and constructive comments on this manuscript by the reviewer. The comments have been very thorough and useful in improving the manuscript. We strongly believe that the comments and suggestions have increased the scientific value of the revised manuscript by many folds. We have taken them fully into account in revision. We are submitting the corrected manuscript with the suggestion incorporated in the manuscript. The manuscript has been revised as per the comments given by the reviewer, and our responses to all the comments are as follows:

Reviewer #1:

Comments and Suggestions for Authors

Journal: International Journal of Molecular Sciences

Manuscript ID: IJMS-1339215

Title: The E3 ubiquitin ligase NEDD4-1 mediated temozolomide-resistant glioblastoma, through PTEN attenuation and redox imbalance Nrf2-HO-1 axis

Authors: Hao-Yu Chuang, Li-Yun Hsu, Chih-Ming Pan, Narpati Wesa Pikatan, Vijesh Kumar Yadav, Iat-Hang Fong, Chao-Hsuan Chen, Chi-Tai Yeh**, Shao-Chih Chiu*

Revision:

Chung et al. present a well-written manuscript which introduces NEDD4-1 mediated PTEN degradation in TMZ-resistant glioma. They found two specific microRNAs which down-regulates in TMZ-resistant glioma cell line and these microRNAs are essential for sustaining PTEN expression via down-regulation of NEDD4-1. In clinical aspect, they have shown that (1) the correlation between NEDD4 over-expression and worse prognosis in gliomablastoma, (2) I3C-induced NEDD4-1 down-regulation enhance TMZ-therapeutic efficacy. There are no major comment.

Minor Comments:

Q1: Reviewer #1: line 117-119. If this manuscript is first report for NEDD4-1 over-expression in GBM, the author should describe it.

A1: We sincerely thank the reviewer for the time taken to review our work, and for the valuable suggestions given. As per the suggestions, we described the NEDD4-1 importance in the main text of our newly edited manuscript:

“To our knowledge, this is the first research on the novel role of NEDD4-1, an E3 ligase in GBM TMZ resistance”. Kindly see Page 6, lines 123-124.  

Q2: Reviewer #1: line 139-140,148-149,160-161,181-182191-192,199-200,204-205,217-218. The author should insert spaces.

A2: We are grateful for the reviewer’s comment. In this revised manuscript, we have rechecked and inserted the spaces where ever is needed as suggested by the reviewer. Please kindly refer to the newly attached manuscript.  

Q3: Reviewer #1: Figure 1C. The author should explain histological color.

A3: We appreciate the reviewer’s suggestions. We agree with the suggestions given, we have rechecked and described the histological color (Hematoxylin: Deep Blue and DAB (3,3-Diaminobenzidine): Dark brown) of Figure 1C. Please kindly see our revised Figure 1C, and main text material methods: Page 14, line 327-329 

Q4: Reviewer #1: Figure 1D. The author should describe P value mean of ***.

A4: We thank the reviewer for the insightful comment, we have described the p-value *** means in the Figure 1 legend.  Please kindly see our revised Figure 1, legend description.  

Q5: Reviewer #1: Figure 1E and 1F. Half of the bottom of P values disappear.

A5: We sincerely appreciate reviewers insightful comments on the figure improvement. In this newly edited manuscript, we have redrawn the figure 1E, F and properly labelled the p-value data, so that it will be appropriately visible. Please kindly see our amended Figure 1E, F.  

Q6: Reviewer #1: Figure 2A and 2D right. Can the author clear the background?

A6: We appreciate the reviewer’s suggestions. We have removed the background from the figure to make it more perceptible. Please kindly see our amended Figure 2A and 2D.  

Q7: Reviewer #1: Figure 2 legend. [resisttant] is miss-spelling.

A7: We appreciate the reviewer for insightful comments. In this revised manuscript, we have proofread all the spellings/typographical errors. Kindly check the Figure 2 legends. 

Q8: Reviewer #1: Figure 4D. The readers will not be able to find cells in this photo. Can the author modulate color balance or draw the line of migrated-tip cells?

A8: The reviewer's insightful comments are greatly appreciated. In this newly edited manuscript, we have redrawn Figure 4D to show the cells in the migration assay results. Please kindly see our amended Figure 4D.  

Q9: Reviewer #1: Figure 4F NESTIN,SOX2. Can the author clear the background?

A9: We appreciate the reviewer’s suggestions. We have removed the background from the western blot image figure to make it more clear. Please kindly see our amended Figure 4F.  

Q10: Reviewer #1: line 337,342. The author should simply explain TBHp and DCFDA.

A10: We thank the reviewer for bringing up this good point, and we agree with this. In this revised manuscript, we have briefly described the TBHp and DCFDA. Please kindly see Page 9-10, lines 214-219 and 222-224 of our result section part in the main text.  

Q11: Reviewer #1: Figure 4B. U251 and U87 are lower expression of NEDD4-1 than each U251R and U87R. Are these cells higher cell viability upon I3C treatment?

A11: We are grateful for the reviewer’s comment. Previously we did not measure the I3C effect on the cell viability of parental cells (U251 and U87), but in this newly edited manuscript, we have rechecked and plotted the effect of I3C on the parental cells. Please kindly see our revised attached Figure 4B and the result section. Kindly see Page 9, line 200-202.  

Q12: Reviewer #1:Figure 6B. Y axis is not "tumor size". Please change to "Body size".

A12: Thanks to reviewers for the valuable comments. We sincerely apologize for our oversight, and we have revised “tumor size” to “Bodyweight” in our Figure 6B Y-axis. Please kindly see our revised attached Figure 6B.

Q13: Reviewer #1:line 385-387. Please check the grammer.

A13: Thank you very much for your valuable comments. We sincerely apologize for these errors, and we have revised our manuscript accordingly; rechecked the grammatical error in the whole manuscript. Please kindly refer to the new attached edited manuscript.

Q14: Reviewer #1:line 399. How mechanism does result in dysregulated miRNA? This concern may be future work and please discuss it if you are possible. Moreover, Dai et al. (Cancer Res.70.2951) have shown that FoxM1B transcription factor is overexpressed in human glioma tissues and upregulated NEDD4-1. Is there no alteration of FoxM1B expression in your glioma cell line?

A14: We are grateful for the reviewer’s comment. Studies have suggested that a complex set of miRNAs is involved in the regulation of the stem-like characteristics in glioblastoma, these miRNA’s can be oncomirs or tumor suppressors. We have reworded the text and discussed a little about the miRNA-CSCs connections.

Please kindly see our revised main text of the manuscript, Page 11-12, line 268-277. Reviewers comments are valuable, but we sincerely apologize that, in this current manuscript, we did not measure the expression of FoxM1B transcription factor in GBM tissue together with the expression of NEDD4-1.

Reviewer 2 Report

The manuscript by Dr. Chiu and his colleague illustrated the important role of NEDD4-1 in the TMZ-resistant GBM cancer. They demonstrated that NEDD4-1 was significantly overexpressed in the TMZ-resistant GBM. This upregulation attenuated PTEN expression and promoted AKT oxidative stress signaling. They found the combination treatment of NEDD4-1 inhibitor with TMZ results in the synergistic effect both in vitro and in vivo. However, it seems still unclear whether NEDD4-1/NRF2/HO-1 signaling axis is involved. There are a few points that need to be addressed:

  1. Please make sure the protein name is consistent. NEDD4-1 or NEDD-4-1?
  2. Fig 1A: please label y axis.
  3. For Fig3: authors stated that miR-3129-5p and miR-199b-3p regulate NEDD4-1 expression in GBM cell lines. But their data fail to show the direct correlation. If NEDD4-1 is knocked down or knocked out in TMX resistant cell lines, will the expression of miR-3129-5p or miR-199b-3p change? Also if NEDD4-1 plays essential role in TMZ-resistant GBM, when NEDD4-1 is knocked down or knocked out in TMX resistant cell lines, will cells become sensitive to TMZ treatment?
  4. For Fig 3B and Fig3C, western blot should be done to quantify the difference. For Fig3C, it’s hard to see the difference in the merged images, please show the single color images as well.
  5. For Fig 4B, what about TMZ sensitive cell lines?
  6. For Fig 4F, author stated “we demonstrated that I3C treatment significantly reduced NEDD4-1 expression, 319 together with the reduction in the expression of stemness markers (NESTIN and SOX2), and EMT (N-cadherin and vimentin) markers”. N-cadherin and vimentin are mesenchymal markers, what about epithelial markers such as E-cadherin or γ-catenin?
  7. For Fig 5A, the authors stated “As expected, the NRF2/HO-1 signaling cascade was also strongly activated in the cells induced with TBHp.” Comparing control and TBHp treatment alone, I don’t think the difference for Nrf2 (1 to 1.4) and HO-1 (1 to 1.6) are statistic significant. Also, I3C treatment alone (without TBHp treatment) should be included.
  8. For Fig 5E, the authors stated “observed the synergistic effect of combination drugs on TMZ-resistant GBM cells” Please calculate and show combination index.
  9. For Fig 6B, the y-axis label is wrong. I guess authors mean “body weight”?

Author Response

Reviewer #2:

Comments and Suggestions for Authors

Q1: Reviewer #2: Please make sure the protein name is consistent. NEDD4-1 or NEDD-4-1?

A1: We sincerely thank the reviewer for the time taken to review our work. In this revised manuscript, we have proofread all the spellings/typographical and grammatical/technical errors, and also make the gene name NEDD4-1 consistent in the allover the main text.

Q2: Reviewer #2: Fig 1A: please label y axis.

A2: We thank the reviewer for the insightful comment, we have labelled the Y-axis of Figure 1A.  Please kindly see our revised Figure 1A.

Q3: Reviewer #2: For Fig3: authors stated that miR-3129-5p and miR-199b-3p regulate NEDD4-1 expression in GBM cell lines. But their data fail to show the direct correlation. If NEDD4-1 is knocked down or knocked out in TMX resistant cell lines, will the expression of miR-3129-5p or miR-199b-3p change? Also if NEDD4-1 plays essential role in TMZ-resistant GBM, when NEDD4-1 is knocked down or knocked out in TMX resistant cell lines, will cells become sensitive to TMZ treatment?

A3: We again thanks the reviewers for the valuable suggestions and for bringing up this good point. We are sorry for the confusion, we have checked the data as shown in Table 1 and Figure 3. Out of 10 miR’s, the bioinformatics analysis of miR’s, hsa-miR-3129-5p and hsa-miR-199b-3p are showing very strong predictive correlation with NEDD4-1. Also, the loss or gain of function of these miR’s validating NEDD4-1 is of the key gene target. Furthermore, the luciferase assay, advocates that these genes are the strong target of the aforementioned miR’s. Regarding the knockdown of NEDD4-1 effect on the resensitization of TMZ-resistant GBM cells toward TMZ, we apologize that we have not checked it, but through our above-mentioned results, we confirm the effect of these key miR’s in regulating the expression of NEDD4-1 and altering these miR’s expression may sensitize these TMZ-resistant GBM cells.

Q4: Reviewer #2: For Fig 3B and Fig3C, western blot should be done to quantify the difference. For Fig3C, it’s hard to see the difference in the merged images, please show the single color images as well.

A4: We appreciate the reviewer’s suggestions. We have quantified all the western blot images to show the effect of treatment, and we also make the figure more clear as per the valuable suggestions from the reviewers. Please kindly refer to our newly amended Figures 3B and Fig3C.  

Q5: Reviewer #2: For Fig 4B, what about TMZ sensitive cell lines?

A5: We are grateful for the reviewer’s comment. Previously we did not measure the I3C effect on the cell viability of parental cells (U251 and U87), but in this newly edited manuscript, we have rechecked and plotted the effect of I3C on the parental sensitive cell lines. Please kindly see our revised attached Figure 4B and the result section. Kindly refer to Page 9, line 200-202. 

Q6: Reviewer #2: For Fig 4F, author stated “we demonstrated that I3C treatment significantly reduced NEDD4-1 expression, 319 together with the reduction in the expression of stemness markers (NESTIN and SOX2), and EMT (N-cadherin and vimentin) markers”. N-cadherin and vimentin are mesenchymal markers, what about epithelial markers such as E-cadherin or γ-catenin?

A6: We thank the reviewer for bringing up this good point, and we agree with this. In this revised manuscript, we have re-run the treatment samples for western blot analysis to measure the expression of E-cadherin an epithelial marker. Please kindly refer to our new western blot image in Figure 4F.  

Q7: Reviewer #2: For Fig 5A, the authors stated “As expected, the NRF2/HO-1 signaling cascade was also strongly activated in the cells induced with TBHp.” Comparing control and TBHp treatment alone, I don’t think the difference for Nrf2 (1 to 1.4) and HO-1 (1 to 1.6) are statistic significant. Also, I3C treatment alone (without TBHp treatment) should be included.

A7: We are grateful for the reviewer’s comment. We have rechecked and rerun the samples a replaced the old western blot image with the new run results, in this current results after quantifying we can see major statistical differences in treated and control samples. Together with the NRF/HO-1 markers we also added the alone effect of  I3C on the expression of the aforementioned markers. Please kindly refer to our new Figure 5A.

Q8: Reviewer #2: For Fig 5E, the authors stated “observed the synergistic effect of combination drugs on TMZ-resistant GBM cells” Please calculate and show combination index.

A8: We are grateful for the reviewer’s comment in bringing up this good point, it will help to strengthen our manuscript. We have calculated the CI index of combination treatment of both the drugs to show the synergistic effect. Please kindly refer to our new added Figure 5E (right) and text on page 10, line 227-229.

Q9: Reviewer #2: For Fig 6B, the y-axis label is wrong. I guess authors mean “body weight”?

A9: Thanks to reviewers for the valuable comments. We sincerely apologize for our oversight, and we have revised “tumor size” to “Body weight” in our Figure 6B Y-axis. Please kindly refer to our revised attached Figure 6B.

Reviewer 3 Report

The paper by Hao-Yu Chuang et al., entitled: “The E3 ubiquitin ligase NEDD4-1 mediated temozolomide-re-sistant glioblastoma, through PTEN attenuation and redox im-balance Nrf2-HO-1 axis”.  This study regards the role of the NEDD4-1 ubiquitin ligase (Neuronal precursor cell-expressed developmentally down-regulated 4-1), that recruits an E2 ubiquitin-loaded enzyme, ultimately catalyzing the transfer of ubiquitin.

In this study, the authors show that NEDD4-1 is overexpressed in GBM tissue samples and in temozolomide (TMZ)-resistant U251 and U87 glioma cell lines. They search for candidate miRNA to NEDD4-1 mRNA and, among these, for down-modulated miRNAs in the TMZ-resistant U87 cells. Two miRNAs (miR-3129-5p and miR-199b-3p) directly modulate NEDD4-1 UTR mRNA and reduce NEDD4-1 protein. Resistant cells, exposed to I3C, inhibiting NEDD4-1, exhibit reduced migration, invasion, and spheres formation. Finally, the authors show that the combination of TMZ and I3C is efficacious even in TMZ-resistant cells, thus opening interesting perspectives.

By ubiquitin targeting for degradation, the NEDD4 E3 ligase negatively regulates tumour suppressors like PTEN, consistently with NEDD4 overexpression in human cancer. Ubiquitylation enhances Akt translocation and activation on the cell membrane. This topic is very relevant, considering that anti-GBM therapies are unsatisfying and finding new regulatory circuitries could provide new targets.

However, the English language is poor, there are many grammar/spelling mistakes. Also, the paper has many weaknesses. More observations follow:

Fig.1 The datasets indicated in Fig.1A (Berchtold, Harris etc) are not further explained in their composition and criteria, neither mentioned in Results nor in Methods. Fig.1B: although the TCGA GBM is available online, the main clinical findings regarding these patients should be mentioned. Were patients studied in Fig.1C and D selected similarly to A and B, with regard to age, gender, tumor grade etc?

Fig.2 The methodology to obtain resistant cells is not well described. How long the cells were kept in 150 micromolar TMZ? Were they cloned or whole cell populations were employed for further studies?

In the legend to Fig.2: “NEDD4-1 is highly expressed in temozolomide-resisttant GBM cancer stem cells.” U87 and U251 are indeed patient-derived GBM cell lines and not cancer stem cells.

Fig.3 The authors show that transfection of miR-3129-5p or/and miR-199b-3p mimics reduce NEDD4-1 and increases PTEN protein levels. However, this does not imply a reduced ubiquitination of PTEN.  To conclude that the underlying mechanism is indeed a NEDD4-1 -dependent decreased targeting of PTEN for degradation (Fig.7), they should test PTEN ubiquitination in a dedicated experiment.

Fig.4 Treatment of TMZ-resistant U87 and U251 cells with Indole-3-carbinol (I3C) results in an approximately 50% reduction of NEDD4-1. Can this account for the strong decrease in cell viability, sphere forming ability, migration and invasion?

Does I3C affect non-resistant U87 and U251?

Fig.5 I3C is known to inhibit extracellular elastase cleavage of CD40, thus affecting NF-kB-dependent cell survival. Several signaling pathways are known to be inhibited by I3C (PI3K-Akt, Wnt, NFkB, IGF-1 etc). How do you know that the effects observed are exclusively due to the down modulation of NEDD4-1 and consequent reduced PTEN ubiquitination? There may be concurrent effects through other pathways.

Do you obtain similar effects by transfecting siRNA to PTEN in the U87 or U251 cell lines?

Few more points:

The last author seems to be missing in the front page.

The English language is very poor and the paper should be re-written. The expressions used are inaccurate or wrong (for example, “The NEDD4-1 gene is an E3-ubiquitin ligase”, the correct would be: “The NEDD4-1 gene encodes an E3-ubiquitin ligase”)

The recent review by Humphreys et al, on “The role of E3 ubiquitin ligases in the

development and progression of glioblastoma” on Cell Death & Differentiation vol 28, p 522–537 (2021) should be cited.

Author Response

Reviewer #3:

Comments and Suggestions for Authors

The manuscript by Dr. Chiu and his colleague illustrated the important role of NEDD4-1 in the TMZ-resistant GBM cancer. They demonstrated that NEDD4-1 was significantly overexpressed in the TMZ-resistant GBM. This upregulation attenuated PTEN expression and promoted AKT oxidative stress signaling. They found the combination treatment of NEDD4-1 inhibitor with TMZ results in the synergistic effect both in vitro and in vivo. However, it seems still unclear whether NEDD4-1/NRF2/HO-1 signaling axis is involved. There are a few points that need to be addressed:

Q1: Reviewer #3: However, the English language is poor, there are many grammar/spelling mistakes.

A1: We sincerely thank the reviewer for the time taken to review our work, and for the valuable suggestions given. In this revised manuscript, we have proofread all the spellings/typographical and grammatical/technical errors; we took the help of "Wallace Academic Editing" for professional English editing provided by our institution. https://www.editing.tw/en/brief-introduction-wallace to polish the language and to make our manuscript better for more clarity as per the reviewer’s kind suggestions.

Q2: Reviewer #3: Fig.1 The datasets indicated in Fig.1A (Berchtold, Harris etc) are not further explained in their composition and criteria, neither mentioned in Results nor in Methods. Fig.1B: although the TCGA GBM is available online, the main clinical findings regarding these patients should be mentioned.

A2: We sincerely appreciate reviewers insightful comments and suggestions for revising the manuscript. We have rechecked and amended the issue related to Fig. 1A datasets information, we have included the datasets information in the material and methods section Please kindly see our updated main text on page 14, lines 313-319.

Q2: Reviewer #3: Were patients studied in Fig.1C and D selected similarly to A and B, with regard to age, gender, tumor grade etc?

A2:  We sincerely thank the reviewer to review our work. In this revised manuscript, we have included the dataset information. This is our in-house GBM patients cohort data (n = 58), the number of patients and clinicopathological details are shown in Table 1. Please kindly see our updated main text on page 14, lines 321-324.

Q3: Reviewer #3: Fig.2 The methodology to obtain resistant cells is not well described. How long the cells were kept in 150 micromolar TMZ? Were they cloned or whole cell populations were employed for further studies?

A3:  We again thanks the reviewers for the valuable suggestions and for bringing up this key point. We are sorry for our oversight, TMZ-resistant GBM cells were generated as per the protocol suggested by A, Yasuto et.al. 2014 [1] with little modification. We have included the detailed process in the material and methods section. Please kindly see our updated main text on page 14-15, lines 336-341.

Q4: Reviewer #3: In the legend to Fig.2: “NEDD4-1 is highly expressed in temozolomide-resisttant GBM cancer stem cells.” U87 and U251 are indeed patient-derived GBM cell lines and not cancer stem cells.

A2: We thank the reviewer for the insightful comment, we have edited the Fig. 2 legends as per the suggestions.  Please kindly see our revised Figure 2 updated legend information, on page 21, lines 490-499.

Q4: Reviewer #3: Fig.3 The authors show that transfection of miR-3129-5p or/and miR-199b-3p mimics reduce NEDD4-1 and increases PTEN protein levels. However, this does not imply a reduced ubiquitination of PTEN.  To conclude that the underlying mechanism is indeed a NEDD4-1 -dependent decreased targeting of PTEN for degradation (Fig.7), they should test PTEN ubiquitination in a dedicated experiment.

A3: We thank the Reviewer for bringing up this good point, and inviting us to clarify this important point. Many studies have reported the oncogenic role of E3 ubiquitin ligase NEDD4-1 in cancers, NEDD4-1 a proton-oncogene that negatively regulates the PTEN via ubiquitination[2-4], overexpression of NEDD4-1 can post translationally suppress PTEN, (a tumor suppressor) in cancers. In our current study, we applied the bioinformatics analysis of miR’s, hsa-miR-3129-5p and hsa-miR-199b-3p are showing very strong predictive correlation with NEDD4-1. Also, the loss or gain of function of these miR’s validating NEDD4-1 is of the key gene target. Furthermore, the luciferase assay, advocates that these genes are the strong target of the aforementioned miR’s. Through miR-3129-5p or/and miR-199b-3p over-expression/knockdown study we observed that it reduces the expression of NEDD4-1, which reverses the inhibitory effect of NEDD4-1 on the expression of PTEN. Hence, we confirm the effect of these key miR’s in regulating the expression of NEDD4-1 and altering these miR’s expression may sensitize these TMZ-resistant GBM cells.

Q5: Reviewer #3: Does I3C affect non-resistant U87 and U251?

A4: We are grateful for the reviewer’s comment. Previously we did not measure the I3C effect on the cell viability of parental cells (U251 and U87), but in this newly edited manuscript, we have rechecked and plotted the effect of I3C on the parental sensitive cell lines. Please kindly see our revised attached Figure 4B and the result section. Kindly refer to Page 9, line 200-202. 

Q6: Reviewer #3: Fig.5 I3C is known to inhibit extracellular elastase cleavage of CD40, thus affecting NF-kB-dependent cell survival. Several signaling pathways are known to be inhibited by I3C (PI3K-Akt, Wnt, NFkB, IGF-1 etc). How do you know that the effects observed are exclusively due to the down modulation of NEDD4-1 and consequent reduced PTEN ubiquitination? There may be concurrent effects through other pathways.

A5: We thank the reviewer for bringing up this good point. The available study by other groups showed, that the key anti-proliferative role of I3C in cancers, especially in human melanoma, I3C triggers the anti-proliferative action through interaction with NEDD4-1 and disrupting the wild type PTEN degradation [5,6]. NEDD4-1 is a key substrate for I3C, loss of or gain of function of NEDD4-1 and treatment with I3C demonstrated that I3C treatment or knockout NEDD4-1 modulates the expression of key genes such as NEDD4-1, p-AKT/AKT, PTEN, and apoptotic markers in human cancers [5]. Therefore, it might be the I3C effect that is causing the modulation aforementioned process.   

Q7: Reviewer #3: Do you obtain similar effects by transfecting siRNA to PTEN in the U87 or U251 cell lines?

A6: We again thanks the reviewers for the comments and valuable suggestions and for bringing up this good point. We are sorry as we haven't performed the siRNA knockdown of PTEN in our current manuscript, as we mainly checked the epigenetic modification, through modulating the expression of miR’s, which give us the clue that the modulation of these miR’s helps to reverse/resensitize these TMZ-resistant GBM cells.

Q8: Reviewer #3: The last author seems to be missing in the front page.

A7: We are grateful for the reviewer’s insightful comment. We have rechecked and put the *asterisk mark in front corresponding author, Dr Shao-Chih Chiu. Please kindly refer to the front page.

Q9: Reviewer #3: The English language is very poor and the paper should be re-written. The expressions used are inaccurate or wrong (for example, “The NEDD4-1 gene is an E3-ubiquitin ligase”, the correct would be: “The NEDD4-1 gene encodes an E3-ubiquitin ligase”)

A9: We are grateful for the reviewer’s comment, we believe the comments will help to strengthen our manuscript. Done the extensive check of our manuscript and took professional help to resolve the issue and also corrected the expression used for NEDD4-1 in our manuscript main text.

References:

  1. Akiyama, Y.; Ashizawa, T.; Komiyama, M.; Miyata, H.; Oshita, C.; Omiya, M.; Iizuka, A.; Kume, A.; Sugino, T.; Hayashi, N., et al. YKL-40 downregulation is a key factor to overcome temozolomide resistance in a glioblastoma cell line. Oncol Rep 2014, 32, 159-166, doi:10.3892/or.2014.3195.
  2. Amodio, N.; Scrima, M.; Palaia, L.; Salman, A.N.; Quintiero, A.; Franco, R.; Botti, G.; Pirozzi, P.; Rocco, G.; De Rosa, N., et al. Oncogenic role of the E3 ubiquitin ligase NEDD4-1, a PTEN negative regulator, in non-small-cell lung carcinomas. Am J Pathol 2010, 177, 2622-2634, doi:10.2353/ajpath.2010.091075.
  3. Wang, X.; Trotman, L.C.; Koppie, T.; Alimonti, A.; Chen, Z.; Gao, Z.; Wang, J.; Erdjument-Bromage, H.; Tempst, P.; Cordon-Cardo, C., et al. NEDD4-1 is a proto-oncogenic ubiquitin ligase for PTEN. Cell 2007, 128, 129-139, doi:10.1016/j.cell.2006.11.039.
  4. Wang, X.; Trotman, L.C.; Koppie, T.; Alimonti, A.; Chen, Z.; Gao, Z.; Wang, J.; Erdjument-Bromage, H.; Tempst, P.; Cordon-Cardo, C., et al. NEDD4-1 is a proto-oncogenic ubiquitin ligase for PTEN. Cell 2007, 128, 129-139, doi:10.1016/j.cell.2006.11.039.
  5. Aronchik, I.; Kundu, A.; Quirit, J.G.; Firestone, G.L. The Antiproliferative Response of Indole-3-Carbinol in Human Melanoma Cells Is Triggered by an Interaction with NEDD4-1 and Disruption of Wild-Type PTEN Degradation. Molecular Cancer Research 2014, 12, 1621-1634, doi:10.1158/1541-7786.Mcr-14-0018.
  6. Quirit, J.G.; Lavrenov, S.N.; Poindexter, K.; Xu, J.; Kyauk, C.; Durkin, K.A.; Aronchik, I.; Tomasiak, T.; Solomatin, Y.A.; Preobrazhenskaya, M.N., et al. Indole-3-carbinol (I3C) analogues are potent small molecule inhibitors of NEDD4-1 ubiquitin ligase activity that disrupt proliferation of human melanoma cells. Biochemical Pharmacology 2017, 127, 13-27, doi:https://doi.org/10.1016/j.bcp.2016.12.007.

Round 2

Reviewer 2 Report

Although authors stated they made changes, the figures in the revised manuscript is the same as the ones in the original manuscript. Please double check the revised manuscript to make sure the revised figures are included. 

Author Response

We thank the reviewer for carefully reading our manuscript and providing valuable comments. We accordingly response the questions raised by the Reviewer as follows:

Point-by-point responses to reviewer’s comments:

Dear Reviewer,

Coauthors and I very much appreciated the encouraging, critical and constructive comments on this manuscript by the reviewer. The comments have been very thorough and useful in improving the manuscript. We strongly believe that the comments and suggestions have increased the scientific value of the revised manuscript by many folds. We have taken them fully into account in revision. We are submitting the corrected manuscript with the updated figure as per the suggestion. The manuscript has been revised as per the comments given by the reviewer, and our responses to all the comments are as follows:

Reviewer #1:

Q1: Comments and Suggestions for Authors

Although authors stated they made changes, the figures in the revised manuscript is the same as the ones in the original manuscript. Please double check the revised manuscript to make sure the revised figures are included.

A1: We would like to thank the reviewer for the thorough reading of our manuscript as well as the valuable comments. We are sorry for our oversight, we have rechecked and updated all the revised figures in the current manuscript as per the suggestions.  Please kindly refer to our updated figure’s.

Reviewer 3 Report

After the suggested revisions, the paper by Chuang et al., is very much improved, but it still contain minor spelling and grammar mistakes to be improved.

Author Response

Reviewer #3:

Comments and Suggestions for Authors

Q1: After the suggested revisions, the paper by Chuang et al., is very much improved, but it still contains minor spelling and grammar mistakes to be improved.

A: We sincerely thank the reviewer for the time taken to review our work, and for the valuable suggestions. In this revised manuscript, we have proofread all the spellings/typographical and grammatical/technical errors; we took the help of "Wallace Academic Editing" for professional English editing provided by our institution. https://www.editing.tw/en/brief-introduction-wallace to polish the language and to make our manuscript better for more clarity as per the reviewer’s kind suggestions.
